# Expression and Characterization of a Cold-Adapted Alginate Lyase with Exo/Endo-Type Activity from a Novel Marine Bacterium *Alteromonas portus* HB161718^T^

**DOI:** 10.3390/md19030155

**Published:** 2021-03-17

**Authors:** Huiqin Huang, Shuang Li, Shixiang Bao, Kunlian Mo, Dongmei Sun, Yonghua Hu

**Affiliations:** 1Institute of Tropical Bioscience and Biotechnology, Hainan Institute for Tropical Agricultural Resources, CATAS, Haikou 571101, China; lishuangyouxiang@126.com (S.L.); baoshixiang@itbb.org.cn (S.B.); mokunlian@163.com (K.M.); 2Hainan Provincial Key Laboratory for Functional Components Research and Utilization of Marine Bioresources, Haikou 571101, China; 3College of Life Science and Technology, Heilongjiang Bayi Agricultural University, Daqing 163000, China; sdmlzw@126.com; 4Laboratory for Marine Biology and Biotechnology, Pilot National Laboratory for Marine Science and Technology, Qingdao 266071, China

**Keywords:** alginate lyase, cold-adapted, exo/endo-type, *Alteromonas portus*, oligosaccharide, antioxidant activity

## Abstract

The alginate lyases have unique advantages in the preparation of alginate oligosaccharides and processing of brown algae. Herein, a gene *alg2951* encoding a PL7 family alginate lyase with exo/endo-type activity was cloned from a novel marine bacterium *Alteromonas portus* HB161718^T^ and then expressed in *Escherichia coli*. The recombinant Alg2951 in the culture supernatant reached the activity of 63.6 U/mL, with a molecular weight of approximate 60 kDa. Alg2951 exhibited the maximum activity at 25 °C and pH 8.0, was relatively stable at temperatures lower than 30 °C, and showed a special preference to poly-guluronic acid (polyG) as well. Both NaCl and KCl had the most promotion effect on the enzyme activity of Alg2951 at 0.2 M, increasing by 21.6 and 19.1 times, respectively. The TCL (Thin Layer Chromatography) and ESI-MS (Electrospray Ionization Mass Spectrometry) analyses suggested that Alg2951 could catalyze the hydrolysis of sodium alginate to produce monosaccharides and trisaccharides. Furthermore, the enzymatic hydrolysates displayed good antioxidant activity by assays of the scavenging abilities towards radicals (hydroxyl and ABTS+) and the reducing power. Due to its cold-adapted and dual exo/endo-type properties, Alg2951 can be a potential enzymatic tool for industrial production.

## 1. Introduction

Alginate is derived from the cell wall and intercellular substance of brown seaweed, accounting for approximately 22–44% (*w*/*w*) of seaweed biomass [1]. It is a kind of linear, water-soluble, and acidic polysaccharide composed of one or both of *β*-d-mannuronic acid (M) and its C-5 epimer *α*-l-guluronic acid (G) [2]. The M and G subunits form polyguluronic acid (poly-*α*-l-guluronate, polyG), polymannuronic acid (poly-*β*-d-mannuronate, polyM), and mixed random polymer (polyGM) in three types of block structures by α/β-1,4 glycosidic bonds [3]. Alginate is widely used in the field of food and pharmaceutical industry owing to its high viscosity and gelling properties. Further, it has attracted attention as a promising marine biomass for the production of biofuels and chemicals in biorefinery applications [4].

Alginate oligosaccharide (AOS) is an oligomer of alginate, which has the characteristics of low relative molecular weight, good solubility, high safety, and high stability. It is reported that AOS has potential applications in agricultural, health product, and medical industries, since it exhibits significant pharmacological, physiological, and biological activities, including antioxidant, immunomodulation, antibacterial, antitumor, anticoagulation, and plant growth-promoting activities [5,6,7].

Alginate lyase is adopted to catalytically degrade alginate into AOS by *β*-elimination under relatively mild and controllable conditions. According to the substrate specificity, alginate lyases can be classified into three groups: polyM-specific lyase (EC 4.2.2.3), polyG-specific lyases (EC 4.2.2.11), and polyMG-specific lyases (EC 4.2.2) that can degrade polyG, polyM, and polyMG blocks of alginate, respectively [8]. Alginate lyases can be classified into endo- and exo-lytic fashions, according to the mode of degradation of alginate. Endolytic enzymes can cleave the glycosidic bonds inside the alginate with unsaturated oligosaccharides as the main products, while exotype ones can degrade oligomeric alginates and alginate polymers into monomers [9,10]. Furthermore, on the basis of their amino acid sequence similarities, the enzymes are generally grouped into seven polysaccharide lyases (PL) families in the Carbohydrate-Active Enzymes (CAZy) database—that is, PL-5, -6, -7, -14, -15, -17, and -18 [11,12]. Alginate lyases from the PL7 family have been widely studied, e.g., NitAly from *Nitratiruptor* sp. SB155-2 [13], lyA from *Isoptericola halotolerans* NJ-05 [14], Aly1281 from *Pseudoalteromonas carrageenovora* ASY5 [15], and MtAl138 from *Microbulbifer thermotolerans* DAU221 [16]. Alginate lyases are widely derived from marine algae; mollusks; and microorganisms (including bacteria, fungi, and viruses), among which marine bacteria are the most common resources. Many kinds of alginate lyase-secreting bacteria have been reported from marine bacteria, such as *Pseudoalteromona*, *Alteromonas*, *Microbulbifer*, *Zobellia*, *Vibrio*, and *Bacillus* [9,17,18,19]. 

In our previous study, a novel alginate lyase-producing strain, *Alteromonas portus*, was obtained from marine sand sample and taxonomically studied [17]. In this work, a gene *alg2951*, encoding alginate lyase Alg2951, was studied for a cloning, expression, and degradation product analysis. The results showed that Alg2951 was a cold-adapted PL7 alginate lyase with significant preference toward polyG. Moreover, Alg2951 was shown to be a NaCl- and KCl-activated enzyme with exolytic and endolytic activity, which released 4-deoxy-L-erythro-5-hexoseulose uronic acid (DEH) and trisaccharide from sodium alginate. In addition, the antioxidant capacity of the obtained AOS was investigated.

## 2. Results

### 2.1. Screening and Identification of Strain HB161718^T^

Based on the screening results by the plate assay and activity determination, strain HB161718^T^ from coastal sand collected from Tanmen Port in Hainan, China showed a significant activity of alginate lyase. Two types of gellation reactions, white-halo and white-ring zone, were observed on the plate, which showed that two types of alginate lyases, polyM-specific and polyG-specfic iyases, were incubated (Appendix A). On a marine agar 2216 (Difco Laboratories, Detroit, MI, USA.) plate, the colonies were creamy, circular, smooth, and 1 to 2 mm in diameter after incubating at 30 °C for three days. Cells were Gram-stain-negative, rod-shaped with a width of approx. 0.8–1.1 μm and a length of 1.2–2.0 μm, and motile with a single polar flagellum (Appendix A). The 16S rRNA gene sequence (1455 bp) of strain HB161718^T^ was sequenced (GenBank No. MG994978), and the phylogenetic analysis showed a close relationship to members of the genus *Alteromonas*. The results of the phenotypic, phylogenetic, and genotypic analyses clearly indicated that the alginate lyase-excreting isolate HB161718^T^ was classified as a novel *Alteromonas* species, for which the name *Alteromonas portus* sp. nov. is proposed. The type strain is HB161718^T^ (= CGMCC 1.13585^T^ = JCM 32687^T^) [17].

### 2.2. Genome Sequencing and Bioinformatics Analysis of Alginate Lyase Alg2951

The draft genome of *Alteromonas portus* HB161718^T^ comprises 32 contigs and 4,543,354 bp in length (GenBank No. SWCO00000000), with a depth coverage of 449×. The scaffold average length and *N50* are 141,979 bp and 453,001 bp, respectively. Genes (3,948) were detected with 3,834 coding DNA sequences, and the DNA G+C content was 44.1%. A total of 143 proteins were matched with the CAZy database, including 39 glycoside hydrolases (GHs), 49 glycosyl transferases (GTs), 34 carbohydrate esterases (CEs), 10 polysaccharide lyases (PLs), and 11 auxiliary activities (AAs). Further analyses revealed that four of the putative PLs, Alg1687 (PL 17, accession number: WP136782264.1), Alg1691 (PL7, accession number: WP136781786.1), Alg2951 (PL7, accession number: WP136782863.1), and Alg3615 (PL7, accession number: WP136783496.1), were involved in alginate degradation. 

The open reading frame (ORF) of the *alg2951* gene consists of 1,608 bp and encodes a 535-amino acid protein. The online analysis results showed that Alg2951 had a calculated molecular weight (Mw) of 60.66 kDa and the theoretical isoelectric point (pI) value of 5.13. It contained an alginate lyase domain of the PL7 family, and the N-terminal 25–43th amino acids were predicted to be a transmembrane region. To further confirm the attribution of Alg2951, a phylogenetic tree was constructed according to the amino acid sequences of Alg2951 and other reported alginate lyases. As shown in Figure 1, Alg2951 was clearly located in the same clade with the PL7 family alginate lyases and formed a distinct branch. Furthermore, Alg2951 showed the highest similarity (59.9%) with the PL7 alginate lyase (from *Pseudoalteromonas* sp. SM0524). The results of the sequence alignment and phylogenetic analysis imply that Alg2951 belongs to the PL7 family and is a novel member.

Furthermore, a multiple sequence alignment was carried out between Alg2951 and eight well-characterized alginate lyases of the PL7 family (Figure 2). The eight alginate lyases included alginate lyase from *Corynebacterium* sp. ALY-1 (Accession: BAA83339.1), *Flavobacterium* sp. UMI-01 (Accession: BAP05660.1), *Klebsiella pneumoniae* (Accession: AAA25049.1), *Persicobacter* sp. CCB-QB2 (Accession: WP_053404615.1), *Pseudomonas aeruginosa* PAO1 (Accession: AAG04556.1), *Sphingomonas* sp. A1 (Accession: BAD16656.1), *Zobellia galactanivorans* Dsij^T^ (Accession: CAZ98266.1), and *Zobellia galactanivorans* Dsij^T^ (Accession: CAZ95239.1). The results showed that Alg2951 contained the typical conserved motifs of the PL7 family, such as “RS/TELRE”, “QIH”, and “YFKAG”, which were related to the substrate combination and catalytic activity. These amino acids were verified by site-directed mutagenesis to bind the alginate substrate or to determine the catalytic activity [20]. For the PL7 family of alginate lyases, the conserved domain of the second region “QIH” means that it was more inclined to degrade polyG, while “QVH” meant that it was more inclined to use polyM as the substrate [21,22]. Thus, Alg2951 may be also a polyG-preferred alginate lyase.

### 2.3. Expression, Purification, and Activity Detection of the Recombinant Alg2951 (rAlg2951)

To detect the enzyme activity of Alg2951, it was expressed in *Escherichia coli* with His-tag and purified by NTA-Ni Sepharose affinity chromatography. The results of the SDS-PAGE analysis showed that the purified protein appeared as a main band that was close to the predicted Mw of 60.66 kDa (Figure 3a). The alginate lyase activity of Alg2951 was tested by the plate assay and ultraviolet absorption method. The results of the plate assay showed that transparent circles were formed (Figure 3b), which suggested that the recombinant protein had alginate lyase activity. The size of the enzymolysis circle reflecteed the strength of the enzyme activity. Taking the empty vector-induced group and the noninduced group as controls, it was determined that the enzyme activity was produced by the induced recombinant protein, and the activity of Alg2951 was higher than that before purification. 

The substrate specificity of Alg2951 was detected by measuring the increased absorbance at 235 nm of the unsaturated uronic acids that were generated from the oligomers via a *β*-elimination reaction. According to the results of the substrate specificity assay, the recombinant enzyme exhibited a higher activity with alginate and polyG than with polyM (Appendix A), and its ability to degrade polyM was only 2.1% of polyG. The result indicates that Alg2951 prefers to depolymerize polyG and is a member of polyG lyase, which is consistent with the previous prediction that Alg2951 is a polyG alginate lyase. Obviously, the recombinant enzyme could significantly act on polyG and sodium alginate. The preference of Alg2951 suggested it could be used for the production of guluronate oligosaccharides from polyG blocks and th epreparation of polyM blocks from sodium alginates via degrading the polyG and polyMG blocks [23,24]. Similar results were reported for alginate lyase from *Streptomyces* sp. A5 [25] and *Pseudomonas* sp. KS-408 [8], which were only specific for polyM or polyG. Meanwhile, some reported alginate lyases possessed a broad substrate specificity, showing activity towards sodium alginate, polyG, and polyM [26,27]. 

### 2.4. Temperature and pH Properties of the rAlg2951

The enzymatic properties of the purified recombinant Alg2951 were investigated. As shown in Figure 4a, Alg2951 exhibited a maximum enzyme activity at 25 °C. At the temperature range of 15–40 °C, Alg2951 manifested over 60% of the highest activity and 45.2% of the highest activity at remained 4 °C, while there was almost no detectable activity at 60 °C. These results suggest that Alg2951 has cold-adapted characteristics. The thermostability of Alg2951 was determined at a temperature ranging from 4 to 60 °C (Figure 4b). Alg2951 was relatively stable at temperatures lower than 30 °C; approximately 100% of the activity was maintained after incubation at less than 30 °C for 0.5 h. As the temperature rose above 30 °C, the activity declined dramatically; approximately 66.5% of the Alg2951 activity remained after being incubated at 40 °C for 0.5 h and was rapidly inactivated as the temperature increased, and the vast majority of the activity was lost above 50 °C. These results indicate that the thermostability of Alg2951 is quite low, further supporting that Alg2951 is a cold-adapted enzyme.

As shown in Figure 4c, the activity of Alg2951 was the highest at pH 8.0 but below 40% of the maximum activity when the pH value was lower than 7.0 or higher than 9.0. The activity of Alg2951 was stable at pH 8.0 but only retained about 40% of the activity at pH 7.0 and pH 9.0 (Figure 4d). Obviously, the pH has a great influence on the enzyme activity, and the acidic environment has more significant influence. It retained over 25% of the initial activity after being incubated at a pH range of 9.0–10.0, while only below 6.2% activity remained under pH 6.0.

Most of the reported alginate lyases showed their highest activity at about 40 °C [27,28,29,30]. In accordance with the other cold-adapted enzymes studied, the cold-adapted alginate lyases generally have lower optimal temperatures, higher low-temperature catalytic activities, and lower thermal stability than the mesophilic homologs [31]. The cold-adapted alginate lyase performed their optimal catalytic activities at less than 35 °C, generally unstable at temperatures higher than 30 °C, and commonly performed more than 50% of the highest activity at 20 °C [31,32]. Compared with the cold-adapted alginate lyases before [33], Alg2951 showed higher activity at 20 °C and possessed better thermostability at temperatures lower than 30 °C. During the catalytic process of cold-adapted alginate lyases, contamination, consumption energy, and inactivation difficulty can be reduced. Therefore, Alg2951 provides a new catalytic tool for potential industrial applications. 

### 2.5. Effects of Ions and Compounds on the Activity of the Recombinant Alg2951

As shown in Figure 5a, the effects of metal ions and compounds (1 mM and 10 mM) on the activity of Alg2951 were detected. At 1 mM, K^+^, Na^+^, and Mg^2+^ displayed a slightly promoted effect on the enzyme activity. However, Mn^2+^ showed some inhibitory effect with 88.3% of the relative activity, followed by Fe^2+^ and Ba^2+^ with 79.0% and 73.3%, respectively. Zn^2+^ and the chelating agent ethylenediamine tetraacetic acid (EDTA) significantly inhibited the Alg2951 activity, resulting in an enzymatic activity below 35%. The addition of the surface-active agent SDS directly led to the loss of enzyme activity. Moreover, it was found that Na^+^ and K^+^ at 10 mM greatly increased the activity of Alg2951, reaching 1.4 and 4.0 times of the control group, respectively. Therefore, in the subsequent experiments, the effects of NaCl and KCl at different concentrations on the activity of Alg2951 were further studied.

In addition, as shown in Figure 5b, in the range of the KCl addition from 10 to 800 mM, the enzyme activity was greatly promoted. At a concentration of 0.2 M, the highest enzyme activity was 19.1 times that in the absence of KCl. NaCl had the same promotion effect on the enzyme activity, and the highest activity was even higher than that of the KCl group, reaching 21.6 times of that of the control group at 0.2 M (Figure 5c). Thus, an appropriate amount of KCl and NaCl can greatly promote the enzyme activity of Alg2951 and boost oligosaccharide production.

Lots of salt-activated alginate lyases were reported, the activities of which were increased by cations such as K^+^, Na^+^, Mg^2+^, and Mn^2+^ [34,35,36]. In this work, Alg2951 was also a salt-activated alginate lyase, the activities of which could be increased by 21.6 and 19.1 times at 0.2-M NaCl and KCl, respectively. The powerful promoting effect of salt increases the feasibility of utilizing this alginate lyase for industrial applications.

### 2.6. Analysis of the Degradation Product

As shown in Figure 6a, the TLC result showed that the alginate degradation products hydrolyzed for 30 min at 25 °C by Alg2951 (the results performed for 60, 90, and 150 min had the same spots, and data were not shown). The results showed that the main reaction products of alginate were monosaccharide and trisaccharide. The final degradation product of Alg2951 for the alginate polymer was also analyzed by ESI-MS in negative-ion mode (Figure 6b). The peak at 175.02 *m*/*z* (ΔDP1 − H)^−^ corresponded to the molecular mass of the unsaturated alginate monosaccharides and its conversion products DEH. The peak at 549.18 *m*/*z* (ΔDP3 + Na − H)^−^ corresponded to the molecular masses of unsaturated alginate trisaccharide. The esults of the TLC and MS revealed that the Alg2951-mediated degradation of alginate produced DEH and trisaccharide. Taking into account the dominance of monosaccharides and trisaccharides in the hydrolysate, it could be inferred that Alg2951 had high efficiency exo- and endo-activity on the alginate. The product distribution of Alg2951 was similar to the other enzymes, which also exhibited dual endo-/exo-type activity. For example, PL17 family enzyme Alg17B from a marine strain BP-2 degraded alginate to produce oligosaccharides with DP of 2–6, and the main product was a monosaccharide [37], the PL17 family enzyme AlgSH17 from *Microbulbifer* sp. SH-1 also degraded alginate to produce a monosaccharide with small amounts of oligosaccharides with DP 2–6 [38], and the PL15 family alginate lyase from *Sphingomonas* sp. MJ-3 possessed endolytic and exolytic activity and generated both oligomers and monomers [10].

In most cases, the endo-type alginate lyases degrade alginate to produce unsaturated alginate oligosaccharides (AOS), while the exo-type alginate lyases degrade alginate or AOS to produce unsaturated monosaccharides, with the final product of 4-deoxy-L-erythro-5-hexoseulose uronic acid (DEH) by nonreductive conversion [39]. In bacteria, DEH can be converted into 2-keto-3-deoxygluconate (KDG) by a reductase, which is metabolized through the Entner-Doudoroff (ED) pathway [40]. There have been many exo-type alginate lyases reported so far—for example, AlgL17 from *Microbulbifer* sp. ALW1 [41], Atu3025 from *Agrobacterium tumefaciens* [39], Alg17C from *Saccharophagus degradans* 2–40 [42], and AlyFRB from *Falsirhodobacter* sp. alg1 [43]. In recent years, exo-type alginate lyase has received widespread attention, because the unsaturated monosaccharide produced by the degradation of alginate can be easily converted into DEH, a promising material for the production of bioethanol and chemicals [44]. Our results provided a new resource of DEH and trisaccharide.

### 2.7. Antioxidant Activity of the Hydrolysates from rAlg2951-Treated Sodium Alginate

Assays of reducing power and scavenging hydroxyl and ABTS radicals were used to assess the antioxidant activity. As shown in Figure 7a, AOS exhibited a concentration-dependent ability to scavenge hydroxyl radicals and showed a maximum activity of 92.34 ± 0.32% when applied at a concentration of 60.0 mg/mL. The activity was less at low concentrations of AOS, but high activity appeared at high concentrations (>50 mg/mL), stronger than the positive control ascorbic acid (Vc). The IC_50_ value of scavenging hydroxyl was 30.93 mg/mL, which was significantly higher than that reported by Yang et al. (4.3 mg/mL) [38] and Zhang et al. (8.7 mg/mL) [15]. The ABTS radical is also used as a substrate to evaluate the free-radical scavenging ability of an antioxidant. In the present work, the oligosaccharide samples were able to scavenge the ABTS free radical even at low concentrations. As shown in Figure 7b, 15-mg/mL AOS exhibited an optimal ABTS radical scavenging activity of 82.2 ± 0.3%; moreover, 10-mg/mL Vc exhibited an optimal activity of 80.3 ± 0.4%. The IC_50_ value of the ABTS radical was 2.17 mg/mL, which was lower than that of hydrolysates produced by alginate lyase Aly1281 (5.65 mg/mL) [15] and slightly higher than the hydrolysis by alginate lyase AlgL17 (1.85 mg/mL) [41]. It is reported that there is a direct relationship between the antioxidant activity and reducing power. The Fe^3+^ reduction ability is an important index of electron-donating activity [45]. In the reducing power assay (Figure 7c), at a concentration of 1–5 mg/mL, the reducing effects of AOS increased from 0.525 to 1.194 at 700 nm with increasing the enzyme concentration. The reducing power improved with the increased concentration of the samples, displaying a linear correlation, which indicated the ability of AOS to reduce ferric ions to ferrous ions. Moreover, at the concentration of 5–20 mg/mL, the reducing power tended to be stable, showing a weak upward trend. In summary, the hydrolysis by Alg2951 resulted in mainly low-molecular-weight AOS (monosaccharides and trisaccharide), which present good scavenging activities of up to >92% and >82% toward hydroxyl and ABTS radicals, respectively. The disadvantage is that the hydroxyl radical scavenging effect can reach the ideal level when the AOS concentration is higher than 50 mg/mL, while only 47.7% at the concentration of 30 mg/mL. The AOS produced by the enzymatic hydrolysis of Alg2951 shows antioxidant activities and has great potential in the high-value processing of seaweed resources.

## 3. Materials and Methods 

### 3.1. Materials, Strains, and Plasmids

Sodium alginate derived from brown seaweed was purchased from Sangon (Shanghai, China). PolyM and polyG (purity > 90%) were purchased from Qingdao BZ Oligo Biotech Co., Ltd. (Qingdao, China). Other chemicals and reagents used in this study were of analytical grade, except that ethanol used to precipitate oligosaccharide was of chromatographic grade. *Escherichia coli* DH5α and BL21 (DE3) were from TransGen Biotech Co., Ltd. (Beijing, China). The pET28a (+) plasmid was used as the expression vector. 

### 3.2. Screening and Identification of Strain HB161718^T^

Sediment samples were collected from Tanmen Port in Hainan, China (110°37′38′′ E, 19°14′13′′ N) in March 2017 and then spread on modified MA, which contained 5-g sodium alginate per liter additionally. Alginate lyase-excreting activity was tested by plate assay using 1-M calcium chloride as the enzyme-producing indicator [46]. Furthermore, the activity of alginate lyase was determined by the ultraviolet absorption method [47]. One unit of enzyme activity was defined as an increase of 0.1 in absorbance per min at 235 nm. Based on the combined phylogenetic relatedness and phenotypic and genotypic features, strain HB161718^T^ was identified by polyphasic taxonomy [17].

### 3.3. Genome Sequencing and Bioinformatics Analysis of the Alginate Lyase 

Genomic DNA was extracted using the bacterial genomic DNA fast extraction kit (Tiangen Biotech, Beijing, China). The draft genome sequence of strain HB161718^T^ was sequenced by the Illumina HiSeq 2500 platform (2 × 150 paired-ends). The de novo genome assembly was performed using SPAdes version 3.5.0. Protein-coding sequences were predicted with Glimmer version 3.02 software, and polysaccharide lyases were predicted using the CAZy (Carbohydrate-Active Enzymes) database [48]. 

The open reading frame (ORF) of the DNA sequence was translated into the corresponding amino acid sequence using ORF finder (https://www.ncbi.nlm.nih.gov/orffinder/ (accessed on 10 May 2020)). The protein domain prediction was performed by Simple Modular Architecture Research Tool (SMART) (http://web.expasy.org/protparam/ (accessed on 10 May 2020)). The signal peptide was analyzed using the SignalP-5.0 server (http://www.cbs.dtu.dk/services/SignalP/ (accessed on 12 May 2020)) [49]. Domain analysis was performed in the SMART Database (http://smart.enbl-heidelberg.de/ (accessed on 12 May 2020)). The theoretical isoelectronic point (pI) and molecular weight (Mw) were predicted online (http://web.expasy.org/protparam/ (accessed on 12 May 2020)). The neighbor-joining phylogenetic tree was generated based on the reported alginate lyases using MEGA version 7.0 [50]. Multiple sequence alignment was performed among the characterized PL7 family alginate lyases using Clustal X (version 2.1) and obtained using Jalview V2.10.5.

### 3.4. Cloning, Expression, Purification, and Activity Detection of the Alginate Lyase Alg2951

The *alg2951* gene from the genomic DNA of strain HB161718^T^ was used to design the primers for PCR amplification. The primers for cloning the *alg2951* gene (the forward primer: 5′-GGATCCATGTTTAAAATAAAAACAACGCCT-3′ and the reverse primer: 5′-AAGCTTATGTTTCCATTCTTCGCTTAG-3′, with the restriction sites of *BamHI* and *HindIII* in the forward and reverse primers underlined, respectively), were designed using primer premier 6 with reference to the *alg2951* gene sequence. The alginate lyase gene was then subcloned into the pET 28a (+) expression vector for heterologous expression. The recombinant *E. coli* BL21 (DE3) harboring the pET 28a (+)/*alg2951* was cultured in LB medium containing 100-μg ampicillin/mL and induced by adding 0.05-mM IPTG at 16 °C for 21 h. After expression, the cells were harvested by centrifugation and sonicated in lysis buffer. The recombinant protein included in the cell homogenate was purified by an NTA-Ni Spharose column and then analyzed by 12% sodium dodecyl sulfate polyacrylamide gel electrophoresis (SDS-PAGE). 

The purified alginate lyase activity was tested by plate assay [46] and determined by the ultraviolet absorption method [47]. The enzyme activity assays of sodium alginate, polyM, and polyG were defined for investigating the substrate specificity. PolyM, polyG, and sodium alginate were dissolved, respectively, in 50-mM Na_2_HPO_4_-NaH_2_PO_4_ buffer (pH 7.0) with 3-mg/mL working concentration and used as the sole substrate in the test. The reactions were initiated by adding the appropriate enzyme and stopped by heating in boiling water for 10 min. The amount of yielded unsaturated uronic acid was monitored by recording the absorbance of the reaction mixture at 235 nm, using sodium alginate as the reference (100%).

### 3.5. Effects of Temperature and pH Properties Activity and Stability

To determine the optimal temperature, the enzyme activity was measured at various temperatures (4 °C, 25 °C, and 20–70 °C at 10 °C increments) and pH 8.0. To test the thermal stability, the enzyme was preincubated at various temperatures and pH 8.0 for 30 min. To determine the optimal pH, the enzyme activity was measured at various pHs (pH 4.0–10.0 at 1.0 increments) at 25 °C. To test the pH stability, the enzyme was preincubated at various pHs, 4 °C for 1 h. The alginate solution as a substrate was prepared in 10-mM buffers with different pH levels (Na_2_HPO_4_-NaH_2_PO_4_, pH 3.0–8.0; Tris-HCl, pH 8.0–9.0; and glycine-NaOH, pH 9.0–10.0). After each treatment, the enzyme activity was monitored by measuring the absorbance at 235 nm. All reactions were performed in triplicate.

### 3.6. Effects of Metal Ions and Compounds on Alg2951 Activity

To determine the metal ions and compounds, the enzyme activity was measured at 1-mM and 10-mM NaCl, KCl, MgCl_2_, BaCl_2_, ZnSO_4_, MnSO_4_, FeSO_4_, ethylenediamine tetraacetic acid (EDTA), and sodium dodecyl sulfate (SDS), respectively, under the optimum temperature and pH conditions. The enzyme activity without the treatment or addition of extra substances was defined as 100%. All reactions were performed in triplicate.

### 3.7. Analysis of Alg2951 Reaction Products

To elucidate the mode of action of Alg2951 toward the alginate, alginate degradation was performed with 10-g/L sodium alginate in 50-mM phosphate buffer (pH 8.0) as a substrate. The reaction was initiated by the addition of purified Alg2951 (5 U) in a 1-mL reaction volume and performed at the optimal temperature of 25 °C for 30, 60, 90, and 150 min. The reaction solution was then inactivated by heating in boiling water for 10 min, and the polysaccharide was precipitated overnight with three times the volume of chromatography grade absolute ethanol. After centrifugation at 10,000 rpm for 15 min at 4 °C, the supernatant was freeze-dried with a vacuum freezer dryer and applied for TLC and ESI-MS analyses of the degraded products of Alg2951. In the TLC analysis of the oligosaccharide product, a TLC plate (Silica Gel 60 F 254, Merck, Darmstadt, Germany) was used. The solvent system was 1-butanol/acetic acid/water (2:2:1, *v*/*v*). The chromatography plate was sprayed with 5% (*v*/*v*) sulfuric acid in ethanol and then heated at 110 °C for 5 min to visualize spots of oligosaccharide. The monomeric sugar, dimer, and trimer standards were used as markers in TLC chromatography. The molecular mass of the main products obtained from the reaction of alginate with Alg2951 for 10 h was further determined via electrospray ionization mass spectrometry (ESI-MS). A total of 2-μL degradation product was loop-injected into the ESI-MS instrument (Bruker Daltonik GmbH, Bremen, Germany) and operated in negative-ion mode with the following settings: calibration dynamics, 2; capillary voltage, 4.00 kV; cone voltage, 20.00 V; source temperature, 150 °C; desolvation temperature, 350 °C; cone gas flow rate, 50 L/h; and desolvation gas flow, 500 L/h [51].

### 3.8. Antioxidant Activity of the Alginate Degradation Products of Alg2951

The freeze-dried AOS powder degraded at 25 °C for 30 min was applied for the antioxidant activity determination as described above. The assay of the ferric-reducing power of the AOS was carried out in accordance with the method described before [51,52]. The absorbance of the mixture was measured at 700 nm using distilled water as the blank. Hydroxyl radical scavenging activity was determined by using the hydroxyl free-radical scavenging capacity assay kit (Solarbio, Beijing, China), and the absorbance was measured at 536 nm. Total antioxidant activity was determined by using the total antioxidant capacity assay kit with a rapid ABTS method (Solarbio, Beijing, China), and the absorbance was measured at 414 nm. Distilled water and Vc were set as the blank and positive control, respectively. Antioxidant abilities were measured with reference to the procedures of the manuals [41].

## 4. Conclusions

In this work, alginate lyase Alg2951 was cloned from a novel marine bacterium *Alteromonas portus* HB161718^T^, expressed extracellularly, and characterized. The cold-adapted Alg2951 performed its highest activity at 25 °C and performed more than 60% activity at 15–40 °C and was relatively stable at temperatures lower than 30 °C as well. Furthermore, it was a PL7 alginate lyase and could be powerfully promoted by sodium and potassium salts at a concentration of 0.2 M. Meanwhile, it specifically degraded polyG and sodium alginate but had almost no activity on polyM. The TLC and ESI-MS analyses indicated that it could hydrolyze sodium alginate to produce DEH and trisaccharide in an exolytic and endolytic manner. Moreover, the degradation products of alginate by Alg2951 exhibited good antioxidant activities by detecting the scavenging hydroxyl, ABTS radicals ability, and reducing power. This study hopefully provides a potential tool for pharmaceutical and industrial applications.

## Figures and Tables

**Figure 1 marinedrugs-19-00155-f001:**
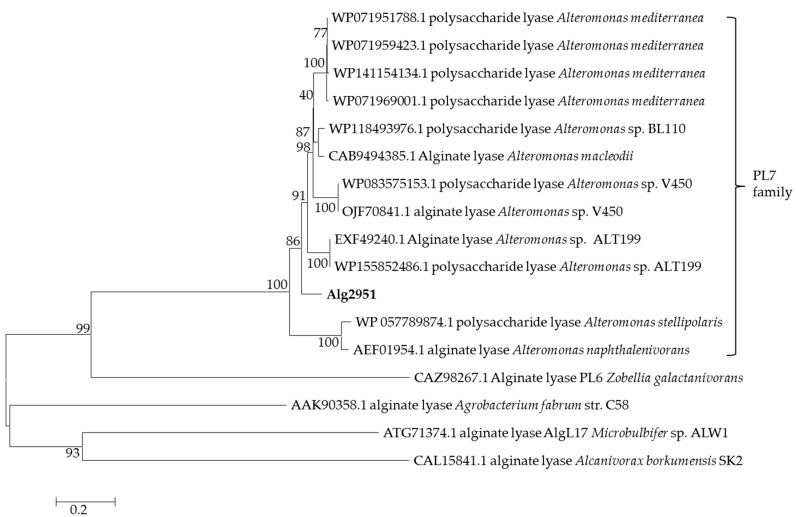
The neighbor-joining phylogenetic tree generated based on the amino acid sequences of the reported alginate lyases by the neighbor-joining method. Bootstrap values of 1000 trials were presented in the branching points.

**Figure 2 marinedrugs-19-00155-f002:**
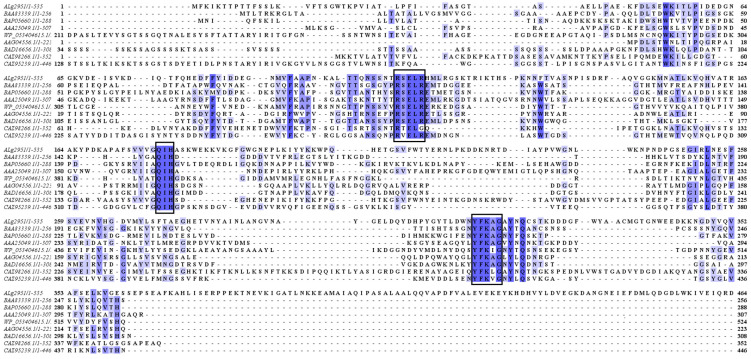
Multiple sequence alignments of Alg2951 and eight well-characterized alginate lyases of the PL7 family. The conserved amino acid regions are marked in the black boxes.

**Figure 3 marinedrugs-19-00155-f003:**
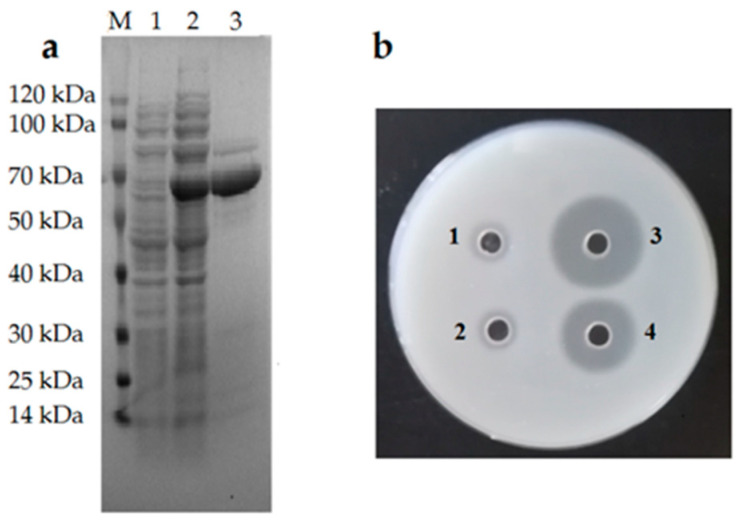
(**a**) SDS-PAGE analysis of the recombinant Alg2951 expression and purification. Lane M, protein marker; lane 1, recombinant protein; lane 2, induced supernatant; and lane 3, the purified Alg2951. (**b**) Detection of alginate lyase activity. (1) Empty vector control, (2) noninduced control, (3) the induced group protein, and (4) the purified protein.

**Figure 4 marinedrugs-19-00155-f004:**
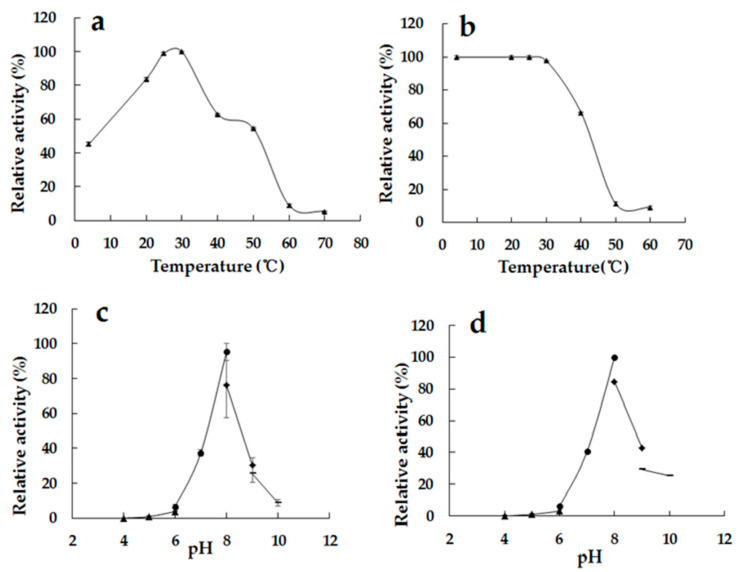
The biochemical characteristics of Alg2951. (**a**) Effect of different temperatures on the activity of Alg2951 (4–70 °C). (**b**) Effect of different temperatures on the stability of Alg2951 (4–60 °C). (**c**) Effect of different pH levels on the activity 2951. (pH 4–10). (**d**) Effect of different pH levels on the stability of Alg2951 (pH 4–10). The highest activity was taken as 100%. Data are given as the means ± standard deviation, *n* =3.

**Figure 5 marinedrugs-19-00155-f005:**
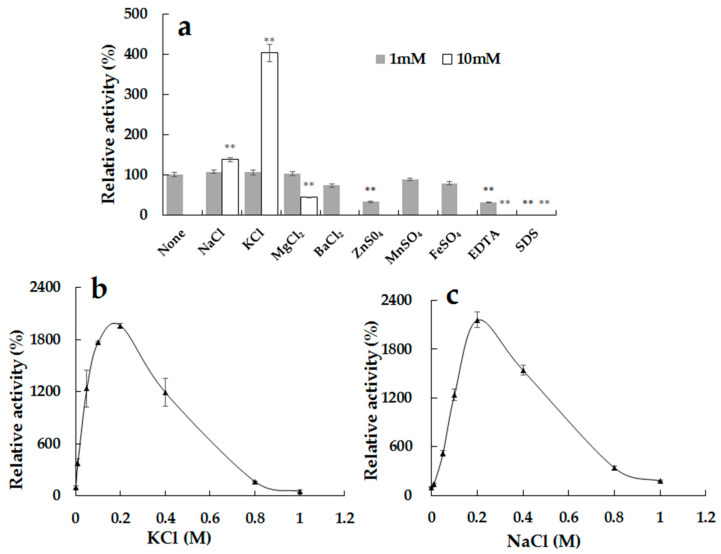
(**a**) Effects of metal ions, ethylenediamine tetraacetic acid (EDTA), and SDS on the activity of Alg2951. ** *p* < 0.01. (**b**) Effect of KCl on the activity of Alg2951. (**c**) Effect of NaCl on the activity of Alg2951. The reaction in the original alginate solution containing no extra substance was taken as 100%. Data are shown as the means ± standard deviation, *n* = 3.

**Figure 6 marinedrugs-19-00155-f006:**
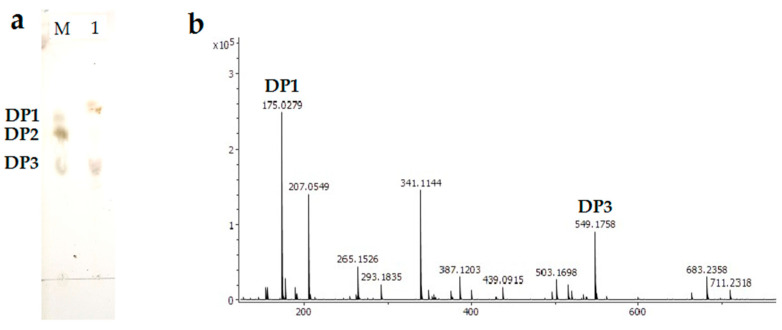
Degradation product analysis of alginate by purified Alg2951 with TLC (**a**) and ESI-MS (**b**). Lane M, the purified monomeric sugar, dimer, and trimer standards and Lane 1, the degradation products performed for 150 min. The DP1 and DP3 peaks represent a monosaccharide and trisaccharide, respectively. DP1 monosaccharide *m*/*z* 175 and DP3 (trisaccharide *m*/*z* 527 + Na^+^ − H: *m*/*z* 549).

**Figure 7 marinedrugs-19-00155-f007:**
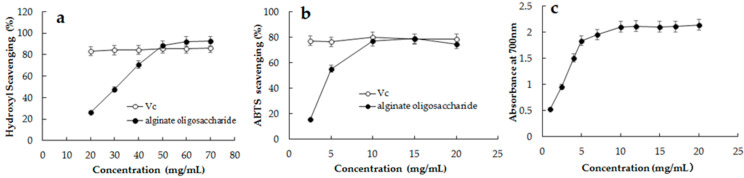
Antioxidant activities of the hydrolysis products from Alg2951-treated sodium alginate. (**a**) Scavenging effect on the hydroxyl radicals. (**b**) Scavenging effect on the ABTS radical. (**c**) Reducing ability. Data represent the mean ± standard deviation of triplicate measurements. Vc is the positive control.

## Data Availability

Not applicable.

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
