# Peer review of "Expression and Characterization of a Cold-Adapted Alginate Lyase with Exo/Endo-Type Activity from a Novel Marine Bacterium Alteromonas portus HB161718T"

_marinedrugs, 2021, doi:10.3390/md19030155_

Round 1
Reviewer 1 Report
This article is of interest as it deals with a new cold-adapted and dual exo/endo-type alginate lyase.
I will agree when authors adjust this work with minor revisions as follows.
(1) Which of the following statements is true?
(a) It also showed special preference to poly-guluronic acid (polyG) at line 23.
(b) alginate lyase with significant preference toward polyM at line 69.
(c) Alg2951 might be a member of polyG lyase at line 155.
(d) Meanwhile it specifically degraded polyM and sodium alginate, but had almost no activity on polyG at lines 412-413.
(2) Line 3 in Figure 3. (a) shows two minor bands above and below a main band at least. So authors can not describe “the purified protein appeared as a single band” at line 136.
(3) It is necessary that the preparation method of AOS applied antioxidant activity.
That’s all.
Reviewer 2 Report
The topic of the manuscript is interesting and relevant due to the great interest in these molecules. Still some sections of the manuscript need to be improved. Furthermore, the long period since the collection of the sediment until the performance of the analysis makes hard to guarantee the confidence of the data and results stated. I am septic about this. Authors should explain how they guarantee the confidence of their data and results knowing the sensitiveness of these analyses and the degradation of the samples that occur along the time.
Specific comments are indicated at the document in attachment. The comments are related to formatting and structure, scientific statements that should be better explained and information should go further in specific sections, and citation of protocols followed should also be well defined.

Round 2
Reviewer 2 Report
In my opinion no further modifications are necessary.